# Integrative Analysis Identifies Candidate Tumor Microenvironment and Intracellular Signaling Pathways that Define Tumor Heterogeneity in NF1

**DOI:** 10.3390/genes11020226

**Published:** 2020-02-21

**Authors:** Jineta Banerjee, Robert J Allaway, Jaclyn N Taroni, Aaron Baker, Xiaochun Zhang, Chang In Moon, Christine A Pratilas, Jaishri O Blakeley, Justin Guinney, Angela Hirbe, Casey S Greene, Sara JC Gosline

**Affiliations:** 1Computational Oncology, Sage Bionetworks, Seattle, WA 98121, USA; 2Childhood Cancer Data Lab, Alex’s Lemonade Stand Foundation, Philadelphia, PA 19102, USA; 3Department of Computer Sciences, University of Wisconsin-Madison, Madison, WI 53715, USA; 4Morgridge Institute for Research, Madison, WI 53715, USA; 5Division of Oncology, Washington University Medical School, St. Louis, MO 63110, USA; 6Sidney Kimmel Comprehensive Cancer Center and Department of Oncology, Johns Hopkins University School of Medicine, Baltimore, MD 21287, USA; 7Neurology, Neurosurgery and Oncology, Johns Hopkins University, Baltimore, MD 21287, USA; 8Department of Systems Pharmacology and Translational Therapeutics, Perelman School of Medicine, University of Pennsylvania, Philadelphia, PA 19104, USA

**Keywords:** neurofibromatosis type 1, nerve sheath tumor, cancer, latent variables, machine learning, supervised learning, transfer learning, random forest, metaVIPER, tumor deconvolution

## Abstract

Neurofibromatosis type 1 (NF1) is a monogenic syndrome that gives rise to numerous symptoms including cognitive impairment, skeletal abnormalities, and growth of benign nerve sheath tumors. Nearly all NF1 patients develop cutaneous neurofibromas (cNFs), which occur on the skin surface, whereas 40–60% of patients develop plexiform neurofibromas (pNFs), which are deeply embedded in the peripheral nerves. Patients with pNFs have a ~10% lifetime chance of these tumors becoming malignant peripheral nerve sheath tumors (MPNSTs). These tumors have a severe prognosis and few treatment options other than surgery. Given the lack of therapeutic options available to patients with these tumors, identification of druggable pathways or other key molecular features could aid ongoing therapeutic discovery studies. In this work, we used statistical and machine learning methods to analyze 77 NF1 tumors with genomic data to characterize key signaling pathways that distinguish these tumors and identify candidates for drug development. We identified subsets of latent gene expression variables that may be important in the identification and etiology of cNFs, pNFs, other neurofibromas, and MPNSTs. Furthermore, we characterized the association between these latent variables and genetic variants, immune deconvolution predictions, and protein activity predictions.

## 1. Introduction

Neurofibromatosis type 1 is a rare disease and a member of the family of RASopathies (diseases caused by germline mutations in genes that encode components or regulators of the Ras/mitogen-activated protein kinase (MAPK) pathway) that occurs in approximately 1 in 3000 patients worldwide and gives rise to cognitive impairment, skeletal abnormalities, and various nerve tumors including gliomas and neurofibromas and is caused by a mutation or deletion in one NF1 allele [1,2,3]. Nerve sheath tumors affect more than 90% of NF1 patients, mostly in the form of cutaneous neurofibromas (cNFs). These tumors grow at the skin surface and can range in number from 10s to 100s of tumors in a given patient [4]. Neurofibromas that occur deeper in the body, including subcutaneous neurofibromas or plexiform neurofibromas (pNFs), occur in 40–60% of NF1 patients and can cause pain and disfigurement among other symptoms [1,4]. Patients with pNFs have a 10% lifetime risk of these tumors developing into malignant peripheral nerve sheath tumors (MPNSTs) which have a 5-year survival rate of 40–50% [5,6]. 

The rise of high-throughput genomic and transcriptomic sequencing has enabled many advances in understanding the molecular etiology of NF1 tumor types [7,8,9,10,11]. Genomic studies of NF1-derived tumors, particularly MPNSTs, have identified key features of tumor growth that could point to potential therapeutic avenues. For example, genomic approaches were recently used to identify the loss of function of polycomb repressor 2 complex components *EED* or *SUZ12* genes, alongside *CDKN2A* and *NF1* gene mutations as crucial co-mediators of MPNST transcriptional dysregulation, pathogenesis, and sensitivity to bromodomain (BRD4) inhibitors [9,11]. Others using genomic approaches to explore nerve sheath tumor biology identified *MET* and *HGF* gene amplifications in MPNSTs. Furthermore, models of *MET*-amplified MPNSTs were subsequently sensitive to the MET inhibitor capmatinib [7]. Transcriptomics-focused approaches have also identified molecular features such as MEK signaling, type 1 interferon signaling, and Aurora kinase A as putative therapeutic targets in NF1 tumors [12,13,14]. Taken together, these and other studies suggest that an integrative approach that combines multiple transcriptomic and genomic datasets might be well poised to identify new therapeutic avenues in MPNSTs and other NF1-related nerve sheath tumors.

Previous genomic profiling studies have demonstrated that many NF1 nerve sheath tumors (MPNST being the exception) are genetically quiet [3,15,16,17] and lack specific signatures that are predictive of drug response. An approach to compensate for the lack of genetic hotspots in NF1 tumors is to focus on combinations of transcriptomic signatures that may be unique to specific tumor types. In other tumor types, transcriptomic landscapes across cancer datasets [18,19] have shown that combining RNA-seq data from similar diseases can identify expression profiles that correlate with prognosis [20], predict drug response [21,22], or identify key tumor biology [23]. 

However, comprehensive analysis of genomic data in NF1 tumors is limited [3]. To enable a larger landscape analysis of NF1 nerve sheath tumors, samples from different studies were collated into a single resource as part of the NF Open Science Initiative, a collaboration between NF-related funding agencies. This resource has been made publicly available through the NF Data Portal, which houses high-throughput data for neurofibromatosis 1, neurofibromatosis 2, and schwannomatosis [24]. 

In this work, we reprocessed and analyzed RNA-seq data from 77 NF1 nerve sheath tumor samples to understand the biological differences that give rise to distinct tumor types in NF1 patients. Given the low sample size compared to the large feature space of RNA-seq data, we applied a transfer learning-inspired approach to meaningfully reduce the feature space with minimal decrease in information content. Transfer learning techniques can leverage large well curated datasets such as recount2 [25] to identify latent variables (LVs)—groups of genes derived from larger repositories of gene expression datasets that exhibit common transcriptomic patterns relevant to a specific subset of samples [26,27]. Although many of these LVs are composed of genes that map to known signatures (i.e., documented in Kyoto Encyclopedia of Genes and Genomes (KEGG) database, and Gene Ontology (GO) consortium database), others are uncharacterized and may allow the detection of novel and meaningful transcriptomic patterns in NF data. As a result, reduction of gene-based expression data to individual latent variables can provide multiple benefits—LVs can highlight differences in known biology in sets of samples, they can uncover previously unknown biology, and they can reduce the impact of technical and experimental differences across multiple datasets [26]. We transferred a machine learning model [27] trained on recount2 to assess LV expression in the NF1 nerve sheath tumor dataset. We then used supervised machine learning with random forests [28] to isolate combinations of such LVs to identify specific molecular signatures that may describe the underlying biology unique to each of the tumor types: cNFs, pNFs, undefined neurofibromas (NFs), and MPNSTs. Finally, we integrated this information with sample-matched variant data, immune cell signatures [29,30,31], and protein activity predictions [32] to provide additional biological context to the most important latent variables. This approach revealed biological patterns that underlie different NF1 nerve sheath tumor types and candidate genes and cellular signatures associated with NF1 tumor heterogeneity. 

## 2. Materials and Methods

### 2.1. Materials Implementation and Data and Code Availability

All analyses were performed using the R programming language. A comprehensive list of packages used and their versions are available in an *renv* lockfile in the GitHub repository. Key packages used include “tidyverse” [33], “PLIER” [26], “synapser” [34], “tximport” [35], “immundeconv” [35], and “viper” [32]. All data analyzed in this article are stored on the NF Data Portal [24] (http://nfdataportal.org) with analyses stored at http://synapse.org/nf1landscape. To recapitulate the analysis from these data, all relevant code can be found at https://github.com/Sage-Bionetworks/NF_LandscapePaper_2019. 

### 2.2. Sequencing Data Collection and Processing

Gene expression data were collected from four independent studies and processed via a workflow at https://github.com/Sage-Bionetworks/rare-disease-workflows/tree/master/rna-seq-workflow to be stored on the NF Data Portal (Table 1). Specifically, raw fastq files were downloaded from Synapse and transcripts were quantified using the Salmon pseudo-alignment tool [36] with Gencode V29 transcriptome. Links to specific datasets and the access teams required to download them can be found using Synapse ID *syn21221980*. 

Genomic variant data were collected from exome-Seq [37] or whole-genome sequencing [38]. Variant call format (VCF) files were processed using “vcf2maf” (https://github.com/mskcc/vcf2maf) according to the workflow located at https://github.com/Sage-Bionetworks/rare-disease-workflows/tree/master/gene-variant-workflow and then uploaded to the NF Data Portal. A list of datasets and the access teams required to download them can be found in Table 2 or syn21266269. 

### 2.3. Latent Variable Calculation and Selection

We analyzed transcriptomic data from NF in the context of latent variables from MultiPLIER, a machine learning resource designed to aid in rare disease analyses [27]. Raw transcriptomic data from NF were retrieved from the NF Data Portal, reprocessed, and stored in Synapse as described above. Salmon output files (quant.sf) files were imported into an R session and converted to HUGO gene names using the “tximport” [35] and “org.Hs.eg.db” [40] packages.

MultiPLIER reuses models that were trained on large public compendia. We retrieved a model [41] that was previously trained on the recount2 RNA-seq dataset [25,42] and then used it to assess the expression of latent variables in the pan-NF dataset. We retrieved code for this analysis from the public repository for MultiPLIER [27] (https://github.com/greenelab/multi-plier). To project the NF data into the MultiPLIER model, we used the GetNewDataB function. This analytical approach is described in more detail in a machine learning training module (https://github.com/AlexsLemonade/training-modules) produced by the Alex’s Lemonade Stand Foundation’s Childhood Cancer Data Lab under an open source license. 

In addition, to ensure orthogonality in the final set of latent variables [43], we calculated the pairwise Pearson correlation of all latent variables by comparing the gene loadings (Appendix A), and identified non-self-correlations greater than 0.5. We eliminated one of each of these highly intercorrelated (Pearson correlation > 0.5) latent variables. This process resulted in a final set of 962 latent variables for further analysis. Code for our implementation of this material is available on GitHub at https://github.com/Sage-Bionetworks/NF_LandscapePaper_2019. 

We used the R function prcomp to compute the principal components for Figure 1A,C both using genes (Figure 1A) and latent variables (Figure 1C).

### 2.4. Generation of Ensemble of Random Forests for Feature Selection 

To select gene expression patterns of interest, we used ensembles of random forests to sufficiently resample our modestly sized dataset. We compared random forest models built using gene expression data as well as latent variables. 

#### 2.4.1. Algorithm Implementation

The main algorithm was implemented using the “caret” and “randomforest” packages in R [44,45]. Appendix A outlines the steps involved in the generation of the ensemble of random forests. Briefly, the full dataset was first split into 80% *model* set and 20% *independent test* set. The function createDataPartition was used to create balanced splits of the data according to the tumor type. We tuned two parameters to the random forest algorithm, *mtrys*, and *ntrees*, using an iterative approach, evaluating *mtrys* values of 1 to 100 and *ntrees* values of 250, 500, 1000, and 2000. We selected the optimal values (*mtry* = 51, *ntrees* = 1000) using fivefold cross-validation, using latent variables as input features. We then split the *model* training set further to generate 500 samples of *training* data (75%) and hold-out *test* data (25%) (balanced splits randomly sampled without replacement). Each of these *training* and *test* datasets were used to train separate random forests to obtain a distribution of F1 scores and feature importance scores (*n* = 500). Given our noisy dataset with limited sample size, the distribution of feature and F1 scores enabled estimation of confidence intervals for the feature importance as well as model performance. 

#### 2.4.2. Feature Selection

The importance of each feature was estimated using raw importance scores that measure the change in correctly classified “class” due to random permutation of the values for the feature. To select the top features for a specific class (i.e., tumor type), we calculated the median *importance score* of each feature from the distribution of *raw importance scores* generated through 500 iterations of random forests. The top 40 features were selected according to the mean decrease of the Gini index. The union of top features from all classes was then used as a restricted feature set to train another 500 iterations of random forests as described above. However, each new forest trained with the restricted feature-set was tested using the *independent test set* to examine the performance of the model on a completely unseen dataset. For each class, the median F1 scores of the new ensemble of forests were compared to the previous ensemble of random forests. Improvement of median F1 scores for each class in the final ensemble of random forests compared to the earlier one suggested that the selected features from each class were sufficiently informative for their classification. This subset of features was then selected for downstream analyses.

### 2.5. Immune Subtype Prediction

To understand the relative immune infiltration across the nerve sheath tumors studied, we used two tumor deconvolution methods: CIBERSORT and MCP-counter, as implemented through the “immunedeconv” R package [29,30,31]. Analysis is located at https://github.com/Sage-Bionetworks/NF_LandscapePaper_2019 and results were uploaded to a Synapse table (syn21177277) that includes the tumor-specific immune cell scores for both algorithms as well as associated tumor metadata.

### 2.6. MetaVIPER

We applied the metaVIPER algorithm [32] to infer protein regulatory activity based on the tissue gene expression profiles. This algorithm builds transcriptional regulatory networks across the cancer genome atlas (TCGA) [32,46] and uses these to build consensus predictions for a sample of other origin. The resulting analysis is uploaded to Synapse and stored at syn21259610 along with tumor-specific metadata. 

### 2.7. VIPER Correlation Clustering and Drug Enrichment Analysis

A heatmap and subclusters of latent variables that had similar VIPER protein predictions were generated using the “pheatmap” R package [47]. We observed five clear clusters of latent variables; these clusters were defined using the R cutree function to isolate the five clusters and their contents. We then calculated the mean correlation of each VIPER protein within each cluster to generate a consensus protein activity prediction for each latent variable cluster. Then, we used gene set enrichment analysis (via the “clusterProfiler” R package [48]) to assess whether drug targets were enriched in the five consensus protein lists. Drug-wise target lists were obtained from the Drug Target Explorer database [49]. Significant enrichment was defined as any positively enriched drug (i.e., a VIPER protein positively correlated with the latent variable cluster) with a Benjamini–Hochberg corrected *p*-value < 0.05. Results were plotted using “ggplot” and “enrichplot” packages [33,50]. To plot the LV cluster expression by tumor type, we calculated the mean expression of all latent variables for each cluster and for each tumor sample.

## 3. Results

We collected mRNA sequencing, exome sequencing, and whole genome sequencing data from previously published or publicly available resources as depicted in Table 3. We applied a combination of methods to identify biological mechanisms of interest in the samples that had patient-derived transcriptomic data. Briefly, we employed a transfer learning-inspired approach to group transcripts into latent variables (LVs), and then selected the LVs that best separate out tumors by tumor type using an ensemble of random forest models. We evaluated the selected LVs for patterns of immune cell gene expression, protein activity, and gene variants.

### 3.1. Pan-NF Transcriptomic Analysis Identified Most Variable Latent Variables in NF1

To measure the diversity of the nerve sheath tumors at the transcriptomic level, we re-processed RNA-seq data from three published datasets [37,38,39] and one unpublished dataset. Despite having four types of nerve sheath tumors (cNFs, pNFs, NFs, and MPNSTs) across four datasets, we observed confounding batch effects (Figure 1A) as some tumor types (e.g., cNF and NF) were derived from separate studies. These batch effects, together with a lack of transcriptomic data from normal tissue from NF1 patients, precluded us from carrying out any meaningful differential gene expression analysis and motivated us to seek alternate approaches to pathway identification.

To reduce the dimensionality of the RNA-seq data and increase the biological interpretability, we applied MultiPLIER, a transfer learning approach trained on the recount2 dataset, an independent dataset comprising thousands of gene expression experiments. This analysis quantified the expression of 962 latent variables (LVs) in all 77 samples using their gene expression data (Figure 1B, Appendix A). We anticipated that as long as technical confounders in our dataset were independent of the technical confounders in the recount2 dataset, we would be able to find some latent variables that were independent of batch effects. We observed that expression of the latent variables was uniformly distributed across the four tumor types (Figure 1B). Furthermore, principal components analysis suggested that this method reduced the size of the previously observed batch effects (Figure 1A,C) as the within-cluster distances were significantly reduced (Appendix A, *p*-value = 2.1 × 10^−182^, Wilcoxon test). 

We then sought to identify which LVs characterize the differences between individual tumors. To find these, we examined the LVs with the 5% largest standard deviation across all LVs. Many of the resulting latent variables (Figure 1D) were found to be associated with known biological mechanisms including metabolic, immune, and transcription-related signatures. However, this analysis did not explore the association between tumor types and latent variables. Therefore, we performed additional analyses to (1) identify the gene expression features that best define each tumor type and (2) elucidate the biological mechanisms underlying the most important latent variables. 

### 3.2. Ensemble of Random Forests Identified Latent Variables That Robustly Describe Individual Tumor Types

To identify these expression features, we used random forest models, an approach that is known for its ability to characterize complex decision spaces better than basic clustering approaches [28]. Random forests achieve this by building decision trees that account for both the values of the features and conditional logic at each “branch” of the tree. Although random forest models are generally used to classify samples, in this study we leveraged the capacity of random forest models to identify features in the data that provide meaningful information. Therefore, in this study, our focus was to identify important features rather than building the most accurate classifier.

We generated a random forest using NF-specific latent variable scores (Appendix A) as described in the Materials and Methods section and depicted in Appendix A. Given the limited number of samples in our dataset, particularly for some tumor types, we anticipated high variability in the performance of our supervised learning model. To obtain a distribution of model performance, we generated 500 training sets (using stratified sampling from the full dataset without replacement), each training their own random forest. We used the distribution of model accuracy measurements and feature importance scores from these forests to estimate overall model performance and feature importance (Figure 2A). We then compared the class-specific accuracy measurements (median accuracy scores: cNF = 0.97, MPNST = 0.67, NF = 0.67, pNF = 0.57) (Figure 2A) to those from forests built with only the top 40 LVs from each class (a total of 98 LVs) from the first forest ensemble. Using the top 40 set of LVs, we were able to improve model performance for each tumor type on an independent test set (median accuracy scores: cNF = 1.00, MPNST = 0.86, NF = 0.86, pNF = 0.75) (Figure 2B). For each tumor type, reducing the feature set to these 98 LVs significantly improved the median accuracy scores for the models (Mood’s median test *p*-values: cNF < 2.2 × 10^−16^, MPNST = 3.789 × 10^−11^, NF = 7.297 × 10^−09^, pNF < 2.2 × 10^−16^). This suggested that the 98 selected LVs with high importance scores for each tumor type were sufficient for characterizing the specific tumor types (Figure 2C). 

### 3.3. Selected Latent Variables Represented Distinct Biology of Nerve Sheath Tumor Types

To further probe the biology underpinning the distinction between NF1 tumor types, we focused on the 98 latent variables selected by the ensemble of random forests, depicted in Figure 3 and listed in Appendix A. We evaluated the tumor-wise LV expression as well as the contributing genes (loadings) for each LV. For example, some latent variables that were selected by the random forest as relevant to predict all four tumor types (Figure 3A) showed differences in expression between various tumor types (dot plots in Figure 3Biii,Ciii), whereas others were less distinct (Figure 3Bi,Ci). By investigating the loadings of each LV, we tried to map them to known biological pathways. For each LV, the loadings of the constituent genes denoted the contribution of that gene to the particular LV. These gene lists associated with the selected LVs can be used to find testable candidates for each tumor type for downstream analysis. For example, one of the four latent variables predictive of all four tumor types was enriched in genes associated with neuronal signaling (Figure 3Bii), suggesting that this measurement is required for the model to distinguish various tumors as the presence of neuronal tissue likely varies across tumor types. Other functionally enriched latent variables, such as those depicted in Figure 3Bi,iii, implicate other biological pathways in tumor growth. However, many of the selected latent variables still remain uncharacterized (few examples shown in Figure 3Ci–iii).

### 3.4. LV Scores May Be Attributed to Specific Gene Variants for Specific Tumor Types

To characterize the 73 LVs with no associated pathway information, we focused on the 40 samples from our dataset (Table 3) that had matched gene variant data (WGS or exome-Seq) to assess if there were any genes that, when mutated, caused a significant change in LV expression. The results of this calculation are found in Appendix A.

We identified 22 latent variables that were significantly (Benjamini–Hochberg adjusted *p* < 0.01) associated with single gene variants (Appendix A). These latent variables, along with the genes whose variants are associated with altered expression, are depicted in Figure 4A. This approach failed to identify any genes that were mutated across multiple tumor types. In the list of genes in which variants were associated with changes in latent variable expression, we identified nine genes with variants in cNFs and two genes with variants in one neurofibroma sample. An example of how these variants are associated with the expression of a latent variable is depicted in Figure 4B. Mutations in the nine cNF-variant genes are associated with lower expression of LV 851. Because most of these variants occur in cNF samples but not in the other tumor types, it is not surprising that this latent variable is down-regulated across all cNFs (Figure 4C). Figure 4D shows that the MAP1B gene, which has been reported to play a role in glioblastoma [51] but has not been studied in NF1-linked tumors, is a major contributor towards this LV. 

### 3.5. Selected Latent Variables Represented Specific Immune Cell Types in the Tumor Microenvironment

Given the limited representation of different NF1 tumor types in our genomic variant dataset, we used alternate gene expression metrics to assess the biological underpinnings of the 73 uncharacterized latent variables. We performed tumor immune cell deconvolution analysis [29,30,31] to identify potential immune infiltration signatures present in the individual tumors using CIBERSORT and MCP-counter (Appendix A). Specifically, CIBERSORT deconvolution indicated the presence of activated mast cells and M2 macrophages in all tumor types (Figure 5A). The analysis further suggested that all of the tumors have a population of resting CD4^+^ memory T cells. The results from MCP-counter, depicted in Figure 5B, show a complementary view of the tumor cell types due to the slightly different categorization of cell types. Specifically, we found the presence of cancer-associated fibroblasts across all tumors that were not captured in CIBERSORT.

We then used the immune scores from CIBERSORT and MCP-counter to probe some of the latent variables selected by the random forest model to better understand their role in NF1 tumor biology. We first searched for tumor deconvolution scores that were correlated with latent variable expression across all tumors. Figure 5C,E show specific LVs where the immune scores of the samples highly correlated with a predicted immune cell type. LV 546, for example (Figure 5C), was found to have a high correlation of activated mast cells in cNF samples. Figure 5E shows that LV 540 has immune scores strongly correlated with T cells in all NF1 tumor types. Complete results of immune scores across all tumors can be found in Appendix A. Together, these results suggest that the selected LVs capture signatures from the tumor microenvironment in NF1 samples.

### 3.6. Selected Latent Variables Captured Protein Regulatory Networks in NF1 Tumors 

Of the 98 latent variables selected by the random forest, 55 could not be characterized via enrichment of the gene loadings (Figure 3), mutational patterns (Figure 4), or immune subtypes (Figure 5). To characterize them, we applied the metaVIPER algorithm [32] to identify specific protein activity measurements that correlated with latent variable expression. This algorithm leverages previously published regulatory network information [52] to infer protein activity in each sample to assign numerical scores of activity across 6168 proteins for each of the 77 samples (Appendix A). We then measured the correlation of these scores and latent variable scores (Appendix A, Figure 6A) to identify functional aspects of the latent variables. 

As seen in previous work [43], we observed that multiple latent variables exhibit similar correlation patterns with active proteins (Figure 6A). We clustered the correlation scores to see if we could group the latent variables with similar protein activity predictions (Figure 6B). In clustering the latent variable–protein activity scores, we observed five distinct clusters of latent variables with similar predicted protein activities (Figure 6B). Aggregation of each cluster of latent variables (mean expression within each cluster) demonstrated that these functional clusters were differentially expressed in the different NF1 tumor types (Figure 6C).

We then assessed the druggability of each of these five clusters by taking the average correlation of each protein within the cluster and performing gene set enrichment analysis against a database of small molecules with known biological targets [49]. This enabled the identification of drugs and drug-like compounds that are significantly enriched for targets in each cluster (Appendix A). For example, cluster 2, which is expressed in pNF, NF, and MPNST more than it is expressed in cNF, has correlated VIPER proteins that are enriched for both clinically approved drugs (dovitinib) and drug-like small molecules (Figure 6D). Furthermore, we found that cluster 3 was enriched for compounds that affect cell cycle progression (e.g., dinaciclib, abemaciclib), whereas clusters 1 and 5 were enriched for compounds such as CUDC-101 (7-[4-(3-ethynylanilino)-7-methoxyquinazolin-6-yl]oxy-N-hydroxyheptanamide) and analogs that inhibit histone deacetylases (Appendix A).

## 4. Discussion

NF1 is the most common of all neurofibromatosis syndromes and is caused by the loss of function of *NF1* gene (a known tumor suppressor) due to mutation or deletion. However, NF1 patients show a great deal of phenotypic heterogeneity [1]. Identification of candidate cellular signaling pathways that differentiate between various tumor types is key for understanding the biology underlying such phenotypic diversity, as well as predicting progression of tumor types towards malignancy. In this study, we integrated various *in silico* resources and analytical techniques to identify candidate genes or pathways unique to different tumor types in an attempt to generate testable hypotheses. By capturing complex gene expression patterns using latent variables (LVs), we identified combinations of LVs that were important to classify tumor types. An ensemble of random forests was then used to select 98 latent variables that were important and sufficient for identifying and classifying the various tumor types with reasonably high accuracy (Figure 2). The selected LVs were then subjected to downstream analyses using different data modalities to gain insight into the composition and relevance of the LVs in the context of NF1 (Figure 3, Figure 4, Figure 5 and Figure 6). The selected latent variables that correlated with known pathways using tumor deconvolution methods confirmed the presence of previously described tumor microenvironment components. Investigation into previously uncharacterized LVs in the context of NF1 suggested the presence of (a) candidate genes for targeted experiments, (b) previously known as well as unknown tumor microenvironment components, and (c) candidate tissue-specific protein regulatory networks for future drug screening experiments. 

To interpret the results of these analyses, we first evaluated the gene loadings of the latent variables that were associated with one or more tumor types. We found that two of the top LVs with lower expression in cNF but higher expression in other tumor types, LV 384 and LV 624 (Figure 3C and Figure 6), had ties to known Schwann cells and NF1 tumor biology, as well as presence of immune cells in the tumor microenvironment. Specifically, the *EGR2* gene, one of the major components of LV 384, has been implicated in diseases associated with the myelin sheath such as Charcot–Marie–Tooth Disease (another disease of the Schwann cell) and is thought to play a role in pathways associated with myelination [53,54]. Indeed, clinical case studies of patients with concurrent NF1-induced tumors and Charcot–Marie–Tooth disease have been reported in the literature, suggesting a possible overlap in underlying biology [55,56,57]. Although the role of *EGR2* in Schwann cell differentiation is still under active investigation [58], *EGR2*-driven pathways have been found to be significantly downregulated in Lats1/2-deficient Schwann cell-based MPNST models [59]. Similarly, the SOCS6 gene, the top component of LV 624, is known to be directly involved in immune signaling via repression of cytokines [60] and has also been found to be a selective tumor suppressor [61]. Its upregulation in the more malignant NF tumors (Figure 3C) suggests that this gene plays a distinct role in NF-related tumors. Additionally, the *RUNX2* gene (a major component of LV 624), has been shown to drive neurofibromagenesis by repression of the *PMP22* gene encoded myelin protein (a Schwann cell component) [62]. The enrichment of such genes in the selected LVs that differentiate nerve sheath tumors from cNFs showcase the importance of Schwann cell biology in these tumors and serves as a proof of principle for our analyses. Furthermore, EGR2 expression in immune cells has been shown to be important for activation of M1/M2 macrophages [63] as well as regulation of CD4^+^ T cells [64]. As discussed later in greater detail, our downstream analysis, using all the selected LVs and tumor deconvolution techniques, identified the enrichment of M2 macrophage and CD4^+^ T cell markers as tumor microenvironment components in our samples (Figure 5A). 

Alternatively, we also evaluated LVs that are uniquely associated with specific tumor types. For example, LV 24 was found to be an important feature for classifying MPNST tumor samples but not the other benign tumor samples (Figure 3Biii). LV 24 was found to be significantly associated with the ΔNp63 pathway (FDR < 0.05), a pathway with implications in determining malignancy and poor survival in various subtypes of pancreatic and squamous cell carcinomas [65]. ΔNp63 signaling in the central nervous system is believed to play a role in neural precursor cell survival and neural stem cell dynamics [66,67], but its role in formation or maintenance of malignant NF1 tumors such as MPNSTs is relatively unexplored. Overall, the evidence surrounding EGR2, SOCS6, RUNX2, and ΔNp63 pathway suggest that LV 384, LV 624, and LV 24 may be promising candidates for experimental follow-up. 

Beyond looking at individual genes that comprise the latent variables, we employed orthogonal algorithms that measured tumor immune activity and regulatory protein activity to identify specific signatures represented by the latent variables. Tumor deconvolution methods (CIBERSORT, MCP-counter) confirmed previously described observations such as the presence of mast cells in NF1 model systems and patient tumors [68,69,70,71,72,73,74]. They also suggested the presence of T cells in human tumor samples. This has previously been described in mouse models of NF1 tumors [75]. In humans, systemic T cell burden has been found to correlate with NF1 nerve sheath tumor progression; T cell presence has also been observed in NF1 gliomas [10,76]. 

Through the metaVIPER protein regulatory activity predictions, we were able to identify putative therapeutic candidates for further evaluation. Specifically, we found two clusters of latent variables expressed in NF, pNF, and MPNST that had regulatory proteins enriched in targets of dovitinib and drug-like small molecules that inhibit receptor tyrosine kinases, as well as cyclin-dependent kinase (CDK) inhibitors such as abemaciclib or dinaciclib. These findings are consistent with previous studies that identify CDK inhibitors as useful in models of *NF1*-deficient or RAS-dysregulated tumors [77,78,79,80]. Because latent variables that are correlated with histone deacetylases (HDACs) were found to be expressed most highly in cNFs (cluster 5, Figure 6C, Appendix A) and MPNSTs (cluster 1, Figure 4A and Figure 6C), compounds such as CUDC-101 and analogs could be potential candidates for treating cNF and MPNST. Indeed, HDAC inhibitors were previously found to be efficacious in in-vitro and in-vivo models of MPNSTs [77,81]. Thus, our results further suggest that this therapeutic approach might be feasible in both MPNSTs and cNFs. Drugs found in other clusters such as dovitinib and lestaurinib in cluster 2, which is expressed most in MPNSTs, pNFs, and NFs (Figure 6C,D), may also merit further study.

Although the present *in-silico* study brings forth various candidate genes and pathways for further follow up, it also presents a few limitations that could be mitigated with future studies, particularly for tumor types with limited samples such as MPNSTs and neurofibromas. Most notably, analyzing genetic variants across tumor types failed to identify relevant variant signatures (Figure 4). This highlights the challenges in variant analyses using samples with limited class representation and motivates our focus on transcriptional signatures. Additional genomic and transcriptomic data from the same biobanks or additional tumor datasets will improve our ability to identify recurrent genetic markers of tumor type. Furthermore, additional data from tumor-adjacent normal tissue would greatly add value to additional analyses on the basis of differential gene expression. Such differential expression analyses were not possible within the scope of this work since these data are not currently available. Additionally, future studies comparing genomic signatures identified here to other publicly available tumor expression and variant data (e.g., the Cancer Genome Atlas or the International Cancer Genome Consortium [19]) may identify genomic similarities between peripheral nerve sheath tumors and other tumors.

## 5. Conclusions

In conclusion, this work proposed a short list of testable hypotheses involving specific biological signatures for NF1-deficient nerve sheath tumors. Verification of these mechanisms in in-vitro and in-vivo models of nerve sheath tumors as well as human NF1 nerve sheath tumor tissue needs active and extensive experimental work. Although we analyzed tumor datasets from four different studies, the addition of other neurofibromatosis-driven tumor datasets will greatly aid identification of commonalities or critical differences to inform therapeutic decisions across the family of neurofibromatoses. This study, together with future work, may guide the repositioning of clinically approved drugs in the context of NF1.

## Figures and Tables

**Figure 1 genes-11-00226-f001:**
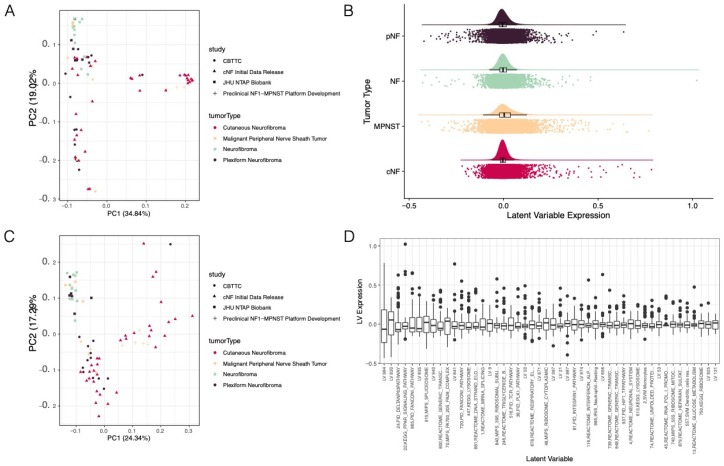
Transfer learning reduced dimensionality and added additional context to gene expression datasets. (**A**) Principal components analysis (PCA) of gene expression data indicated that counts-level data may have been batch confounded. (**B**) The relative distributions of latent variable expression across the four tumor types using a density plot indicated that the majority of latent variables (LVs) had an expression value near 0 and that the four tumor types had similar latent variable expression distributions. (**C**) PCA of LVs indicated that batch effects, although reduced, may still have existed in the LV data (**D**) A look at the 5% most variable LVs across the cohort of gene expression data indicated that the latent variables represented a wide swath of biological processes, as well as some LVs that had no clear association to a defined biological pathway.

**Figure 2 genes-11-00226-f002:**
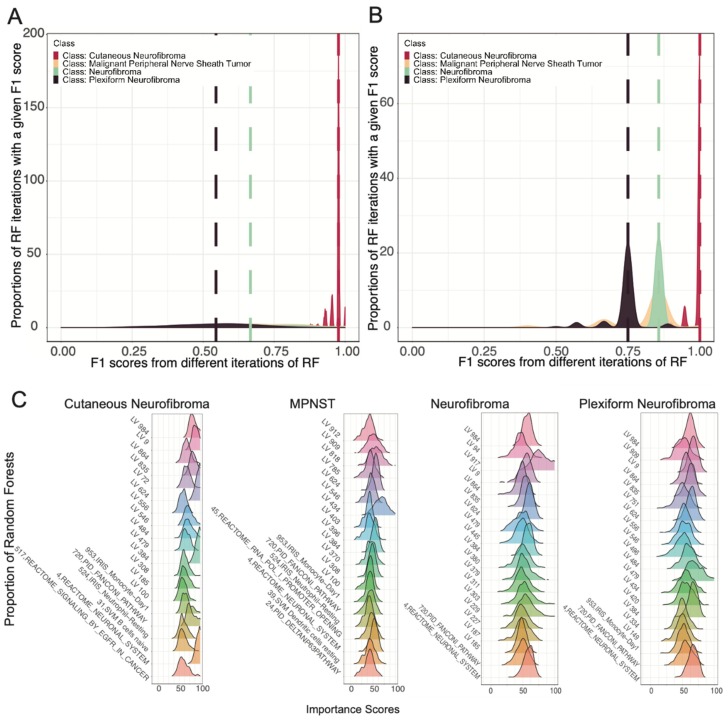
An ensemble of random forests selected the most important latent variables for classifying different tumor types in NF1. (**A**) Density plot showing the distribution of F1 scores of 500 iterations of independent random forest models using all latent variables. (**B**) Density plot showing the distribution of F1 scores of 500 iterations of independent random forest models trained using only the top 40 features with high importance scores for each class obtained from models included in (**A**). (**C**) Ridgeplots of top 20 latent variables selected by the random forest for each tumor type and their importance scores for each class that were selected for later analyses.

**Figure 3 genes-11-00226-f003:**
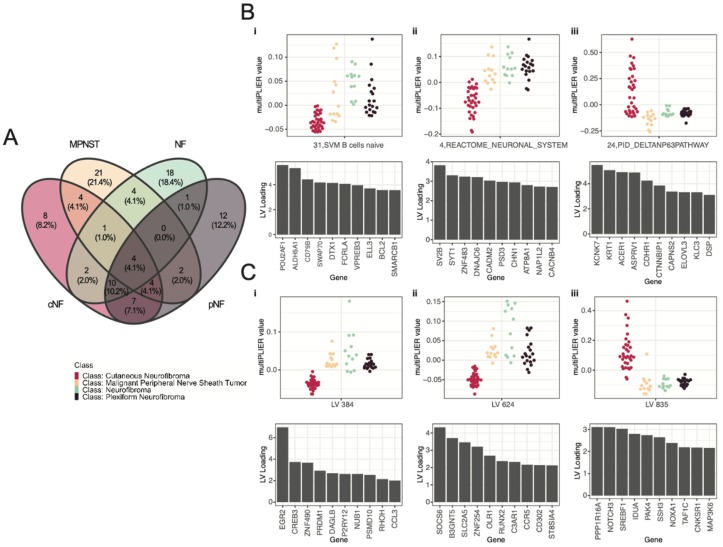
Selected latent variables (LVs) represented gene combinations unique to each tumor type. (**A**) Venn diagram showing the distribution of the top 40 LVs from each tumor type. (**B**,**C**) Total values of the LVs as measured by multiPLIER across samples are represented in the dot-plots, where color of the dots represents the tumor type (“Class” label colors described in the lower left). Loading values for the top 10 genes for each LV are represented in bar-plots below. The higher the loading, the greater impact that the gene expression had on the total multiPLIER value. (**B**,**i**–**iii**) Genes constituting the latent variables associated with known cell signaling pathways. (**C**,**i**–**iii**) Genes constituting the uncharacterized latent variables.

**Figure 4 genes-11-00226-f004:**
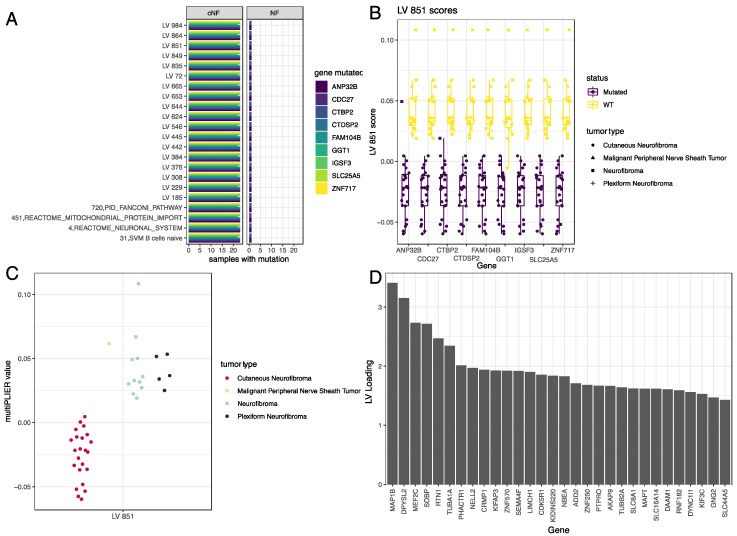
Some genes significantly distinguished expression of latent variables. (**A**) Latent variables (*y*-axis) whose values are significantly altered by mutations in specific genes. (**B**) MultiPLIER value of LV 851 across tumor samples. (**C**) MultiPLIER value of LV 851 across all samples. (**D**) Loading values of the top 20 genes that comprise LV 851.

**Figure 5 genes-11-00226-f005:**
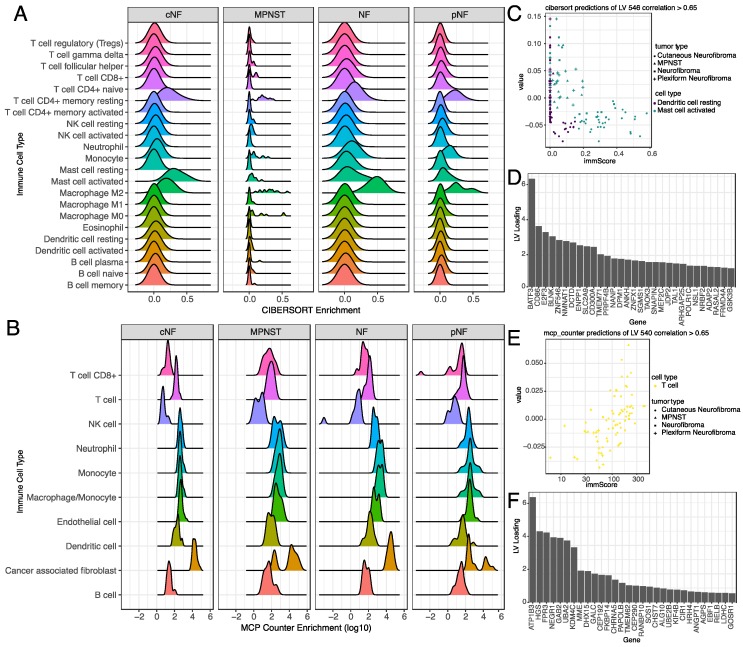
Various immune cell signatures correlated to specific LVs that differentiate tumor types in NF1. (**A**) CIBERSORT deconvolution of bulk nerve sheath tumor expression data predicted the presence of activated mast cells and M2 macrophages and resting CD4^+^ memory T cells in all of the tested tumor types. (**B**) MCP-counter based deconvolution of bulk nerve sheath tumor expression data predicted the presence of cancer-associated fibroblasts across all tumor types, and diversity in T cell population across tumor types. (**C**) Correlation of CIBERSORT immune score (*x*-axis) with expression of latent variable 546 highlighted the increased presence of activated mast cells and resting dendritic cells in cNFs (circles). (**D**) Top 20 gene loadings of LV 546. (**E**) Correlation of MCP-counter score of Tell infiltration (*x*-axis) with LV 540. (**F**) Top 20 gene loadings of LV 540.

**Figure 6 genes-11-00226-f006:**
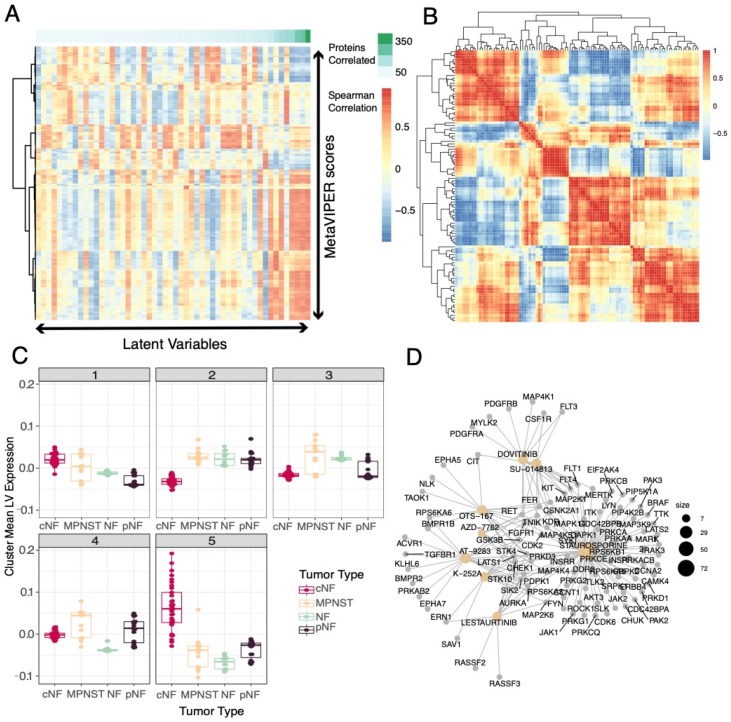
Integration of protein activity information with LVs can identify candidate drug targets for different NF1 tumor types. (**A**) A heatmap of correlation scores of known proteins with regulatory networks (or regulons) that are represented in the characterized and uncharacterized LVs selected above. The green bar across the top depicts how many protein activity scores had a Spearman correlation greater than 0.65. (**B**) Clustering of the LV-correlated VIPER proteins highlighted five clusters of latent variables with similar VIPER protein predictions, suggesting that these five clusters may have functional overlap. (**C**) Mean LV expression within the clusters highlighted differential expression within the clusters across tumor types. Tumor type is indicated by colors on the right. (**D**) Drug set enrichment analysis of the average VIPER protein correlation of cluster 2 identified some drugs and preclinical molecules that are enriched with targets in this cluster.

**Table 1 genes-11-00226-t001:** Description of the gene expression datasets used in the present article.

Dataset Name	Synapse Project Name	Synapse Table Name	Synapse Access Team
WashU Biobank	Preclinical NF1-MPNST Platform Development (*syn*11638893)	WashU Biobank RNA-seq data	WUSTL MPNST PDX Data Access
JHU Biobank [37]	A Nerve Sheath Tumor Bank from Patients with NF1 (*syn*4939902)	Biobank RNASeq Data	JHU Biobank Data Access
cNF Patient Data [38]	Cutaneous Neurofibroma Data Resource (*syn*4984604)	cNF RNASeq Counts	CTF cNF Resource Data Access Group
CBTTC Data [39]	Children’s Brain Tumor Tissue Consortium (*syn*20629666)	CBTTC RNASeq Counts	CBTTC Data Access Group

**Table 2 genes-11-00226-t002:** Description of the genomic variant datasets used in the present article.

Dataset Name	Assay	Synapse Table Name	Synapse Access Team	Synapse Project
JHU Biobank Exome-Seq Data	exomeSeq	Biobank ExomeSeq Data	JHU Biobank Data Access	A Nerve Sheath Tumor Bank from Patients with NF1
cNF WGS Data	wholeGenomeSeq	cNF WGS Harmonized Data	CTF cNF Resource Data Access Group	Cutaneous Neurofibroma Data Resource

**Table 3 genes-11-00226-t003:** Summary of individuals and samples for the NF1 nerve sheath tumors used in this study. All samples have gene expression data and a subset have genomic data derived from whole-exome sequencing or whole genome sequencing. Some neurofibromas did not have more specific pathologic subtyping information available, and therefore were classified as “undefined neurofibromas” or NFs.

Tumor Type	Individuals	Samples	# with Genomic Variant Data
Cutaneous Neurofibroma (cNF)	11	33	23
MPNST	13	13	1
Undefined Neurofibroma (NF)	12	12	11
Plexiform Neurofibroma (pNF)	19	19	5

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
