# Peer review of "Integrative Analysis Identifies Candidate Tumor Microenvironment and Intracellular Signaling Pathways that Define Tumor Heterogeneity in NF1"

_genes, 2020, doi:10.3390/genes11020226_

Round 1
Reviewer 1 Report
By re-analyzing RNA-seq and genomic data from 77 NF1 nerve sheath tumor samples, the authors identify candidate tumor microenvironment and intracellular signaling pathways that define tumor heterogeneity in NF1. The authors apply an in silico statistical and machine learning method, identifying subsets of latent gene expression variables likely specific for cNFs, pNFs, other neurofibromas, and MPNSTs. They also characterized the association between these latent variables and genetic variants, immune deconvolution predictions, and protein activity predictions.
In my opinion, this is a very interesting and well conducted study, but the major criticisms are related to limited number of samples analyzed (although this point is clear to the authors and discussed) and the absence of biological evidence supporting their conclusions. However, it looks like a good starting point for future studies.
Major concerns:
Concerning RNAseq and genomic data, about 50% of them was from cNFs. In table 3, the authors report 33 cNF samples (23 with genomic data) from 11 individuals. I presume that 3 samples were collected from each individual. In my opinion, the dataset does not seem to be sufficiently homogeneous among the different tumor types. This point is not discussed and should be clarified to the reader. Related to this point, in characterizing LVs with no associated pathway information (section 3.4), authors matched gene variant data (WGS or exome-Seq) to assess if there were any genes that, when mutated, caused a significant change in LV expression. They identified 22 latent variables significantly associated with single gene variants. Among these genes, nine were identified in cNFs. Since most of these variants occur in cNF samples but not in the other tumor types, the authors concluded that it is not surprising that the latent variable LV851 is down-regulated across all cNFs. Is it possible that this result is dependent on a non-homogeneous set of cNF samples analyzed? I'd like this point will be clarified.
In the "Discussion" section, the authors state "We foud that two of the top LVs with lower expression in cNF but higher expression in other tumor types, LV384 and LV624 (Figure 3C, Figure 6), had ties to known Schwann cells and NF1 tumor biology". Based on the bar chart in Fig. 3C, I understood because they focused their subsequently discussion on EGR2 as major component of LV384, but I didn't understand why they focused on RUNX2 but not on SOCS6 for LV624. I'd like authors clarify this point.
Minor concerns:
Page 8 (line 279): “(Figures 3Bi and 3i)” should be “(Figures 3Bi and 3Ci)”.
Author Response
Dear Reviewer 1,
We thank you for taking the time to provide feedback for this manuscript and helping us improve it. We are submitting a revised manuscript after incorporating the relevant suggestions. We are also providing point-by-point responses to your questions below:
Major concerns:
Concerning RNAseq and genomic data, about 50% of them was from cNFs. In table 3, the authors report 33 cNF samples (23 with genomic data) from 11 individuals. I presume that 3 samples were collected from each individual. In my opinion, the dataset does not seem to be sufficiently homogeneous among the different tumor types. This point is not discussed and should be clarified to the reader. Related to this point, in characterizing LVs with no associated pathway information (section 3.4), authors matched gene variant data (WGS or exome-Seq) to assess if there were any genes that, when mutated, caused a significant change in LV expression. They identified 22 latent variables significantly associated with single gene variants. Among these genes, nine were identified in cNFs. Since most of these variants occur in cNF samples but not in the other tumor types, the authors concluded that it is not surprising that the latent variable LV851 is down-regulated across all cNFs. Is it possible that this result is dependent on a non-homogeneous set of cNF samples analyzed? I'd like this point will be clarified.
Regarding the first point, we agree with the reviewer that more and class balanced data would improve the results. However, we are constrained by the available data. We acknowledge and address this point in the manuscript in the discussion section: “ Most notably, analyzing genetic variants across tumor types failed to identify relevant variant signatures (Figure 4). This highlights the challenges in variant analyses using samples with limited class representation and motivates our focus on transcriptional signatures. Additional genomic and transcriptomic data from the same biobanks or additional tumor datasets will improve our ability to identify recurrent genetic markers of tumor type. ” We agree that this is likely due to the fact that the variants were only observed in the cutaneous neurofibroma samples. This is addressed in the current manuscript in the Results section corresponding to this genomic analysis: “Since most of these variants occur in cNF samples but not in the other tumor types, it is not surprising that this latent variable is down-regulated across all cNFs (Figure 4C). “ Thus we believe that we have acknowledged and addressed the reviewer’s concern.
In the "Discussion" section, the authors state "We foud that two of the top LVs with lower expression in cNF but higher expression in other tumor types, LV384 and LV624 (Figure 3C, Figure 6), had ties to known Schwann cells and NF1 tumor biology". Based on the bar chart in Fig. 3C, I understood because they focused their subsequently discussion on EGR2 as major component of LV384, but I didn't understand why they focused on RUNX2 but not on SOCS6 for LV624. I'd like authors clarify this point.
We appreciate the reviewer’s attention to this point. There were a couple of reasons for focusing on RUNX2 from LV624 instead of SOCS6. Firstly, we wanted to focus on common threads between LVs which may reinforce support for relevant hypotheses. In this case, the finding of Schwann cell-related proteins in different LVs associated with nerve sheath tumors helped us prioritize Schwann cell-based hypotheses and also helped us contextualize our findings with previous literature in the field. At the same time, we wanted to highlight that looking at multiple genes constituting each LV instead of only the top-loading one may be beneficial during hypothesis generation. However, we acknowledge that SOCS6 is important and added relevant discussion about this. Incidentally, we found multiple genes (in addition to SOCS6), in multiple LVs referring to the involvement of the immune signaling pathways, which prompted us to focus on immune cell biology in these tumors. We have dedicated a separate section in our discussion to elaborate on such involvement in more detail.
We have revised the discussion to clarify these points to the readers.
Minor concerns:
Page 8 (line 279): “(Figures 3Bi and 3i)” should be “(Figures 3Bi and 3Ci)”.
We have revised our manuscript to incorporate this change.
Reviewer 2 Report
This manuscript presents a rather innovative method to re-analyze RNA-seq data in the NF1 context. While the approach used is interesting, the manuscript may reach wider audience expanding some information and with more detailing as specified below.
Major comments:
At line 42 authors introduce neurofibromas that in the following line they specify include subcutaneous neurofibromas or plexiform neurofibromas (pNFs).
From this point on, authors always refer to 4 classes of tissues/samples: cutaneous neurofibromas (cNFs), malignant peripheral nerve sheath tumors (MPNSTs), plexiform neurofibromas (pNFs) and neurofibromas. Therefore, they should make clear starting from first time referring to, if neurofibromas are corresponding to subcutaneous neurofibromas only, and if this is the case maybe an abbreviation such as scNFs could be used in the rest of the manuscript.
In methods (as an example at line 150) and in figures (figure 2, figure S2) they describe "importance scores". Definition of importance scores could help in following the flow of results.
Line 78: the concept of latent variables (LVs) can be exploited better
Line 275: the authors focused on 98 LVs, however in figure 3A a total 94 LVs can be counted
Line 283: authors write that the LV 4.reactome neuronal system is predictive of all four tumor types thus suggesting that the presence of neuronal tissue varies across tumor types. I do not quite understand the meaning of this sentence.
Line 287: Figure 3 Ci-iii, top 3 of what? If authors refer to LVs (as it appears) please add it. Further, I can track LV384 and 624 in figure 2C but I don’t see LV835 in MPNST. Also, related to figures 3B and 3C, why the scaling of Y-axis is different in the three panels? Also, the LV loading Y axis is not with the same scaling/division in panels, and what kind of measure it represents?
In figure 4D (and all figures with “loadings”) is the loadings a total or a partial representation? Again, on the Y-axis LV loading what kind of measure it represents? This should be specified somewhere.
Line 320: in the CIBERSORT deconvolution analysis authors describe resting mast cells and M2 macrophages as present in all tumor types. However, from the figure 5A I don’t see significant presence of mast cell resting (mast cell activated a bit I would say).
Line 323: they describe here, and in figure 5B legend, the presence of fibroblast but in figure 5B (Y-axis) cancer associated fibroblast are represented.
Line 367: It would help highlighting in some way the 5 distinct clusters they refer to in figure 6B. In figure 6C legenda of the x-axis is present in 3 out of 5 panels. Either remove from all panels or put legenda inside the graphs for each panel (considering space constraint)
Line 469: in the conclusions, the last sentence is a bit ambitious. It can be attenuated.
Minor comments:
Is Benjamini-Hochberg. Correct throughout the manuscript.
Data is a plural noun so it should be followed by plural verbs. Check throughout the manuscript (see line 96, 108 as an example)
Line 37: I wouldn’t define NF1 a neuropathic syndrome. Rather is commonly defined as a neurocutaneous syndrome or one of the phakomatoses or RASopathies.
In supplementary figure S2, the panel B (ii) is cut
Reference [39] pages are missing
Author Response
Dear Reviewer 2,
We thank you for taking the time to provide feedback for this manuscript and helping us improve it. We are submitting a revised manuscript after incorporating the relevant suggestions. We are also providing point-by-point responses to your questions below.
Major comments:
At line 42 authors introduce neurofibromas that in the following line they specify include subcutaneous neurofibromas or plexiform neurofibromas (pNFs).
From this point on, authors always refer to 4 classes of tissues/samples: cutaneous neurofibromas (cNFs), malignant peripheral nerve sheath tumors (MPNSTs), plexiform neurofibromas (pNFs) and neurofibromas. Therefore, they should make clear starting from first time referring to, if neurofibromas are corresponding to subcutaneous neurofibromas only, and if this is the case maybe an abbreviation such as scNFs could be used in the rest of the manuscript.
In the initial manuscript we did not clearly describe the nature of the samples in the “NF” class. We’ve added information to the beginning of the Results section and to the Table 3 legend to clarify that these are simply NFs without more detailed clinical metadata; that is, there is no pathologic data available for these samples specifying more information than “neurofibroma.”
In methods (as an example at line 150) and in figures (figure 2, figure S2) they describe "importance scores". Definition of importance scores could help in following the flow of results.
We have revised our manuscript to clarify the concept of importance scores for better readability.
Line 78: the concept of latent variables (LVs) can be exploited better
We have reorganized and added some additional rationale to this section to better explain the putative benefits of using latent variables over gene-wise expression data.
Line 275: the authors focused on 98 LVs, however in figure 3A a total 94 LVs can be counted
We have revised the figure to account for this.
Line 283: authors write that the LV 4.reactome neuronal system is predictive of all four tumor types thus suggesting that the presence of neuronal tissue varies across tumor types. I do not quite understand the meaning of this sentence.
We clarified this statement to read: ”suggesting that the presence of neuronal tissue is required for the model to distinguish various tumors as the presence of neuronal tissue likely varies across tumor types”
Line 287: Figure 3 Ci-iii, top 3 of what? If authors refer to LVs (as it appears) please add it. Further, I can track LV384 and 624 in figure 2C but I don’t see LV835 in MPNST. Also, related to figures 3B and 3C, why the scaling of Y-axis is different in the three panels? Also, the LV loading Y axis is not with the same scaling/division in panels, and what kind of measure it represents?
We removed the ‘top 3’ from the Figure 3 description as it seemed unnecessarily confusing. LV 835 is not in Figure 2C for MPNSTs because it was not selected by the random forest model. We added text in the legend to clarify this. We also added language in Figure 3 to clearly describe the y-axes of plot: “(B, C) Total values of the LVs as measured by multiPLIER across samples are represented in the dot-plots, where color of the dots represents the tumor type (“Class” label colors described in the lower left). Loading values for the top 10 genes for each LV are represented in bar-plots below. The higher the loading the greater impact that the gene expression has on the total multiPLIER value”
In figure 4D (and all figures with “loadings”) is the loadings a total or a partial representation? Again, on the Y-axis LV loading what kind of measure it represents? This should be specified somewhere.
We acknowledge the reviewer's concern and agree that the previous figure legend was a bit ambiguous. We added more detail to the figure legend to clarify that the Y-axis is a measure of the gene loading for the latent variable for the top 20 genes.
Line 320: in the CIBERSORT deconvolution analysis authors describe resting mast cells and M2 macrophages as present in all tumor types. However, from the figure 5A I don’t see significant presence of mast cell resting (mast cell activated a bit I would say).
This was an error; thank you for identifying this. The description of the results have been corrected to read “activated mast cells.”
Line 323: they describe here, and in figure 5B legend, the presence of fibroblast but in figure 5B (Y-axis) cancer associated fibroblast are represented.
We updated this to read ‘cancer-associated fibroblast’.
Line 367: It would help highlighting in some way the 5 distinct clusters they refer to in figure 6B. In figure 6C legenda of the x-axis is present in 3 out of 5 panels. Either remove from all panels or put legenda inside the graphs for each panel (considering space constraint)
We have updated the x-axis labels to be reflected in each of the 5 cluster panels.
Line 469: in the conclusions, the last sentence is a bit ambitious. It can be attenuated.
We have changed the final sentence (“This study, together with future work, will guide the repositioning of clinically approved drugs in the context of NF1.”) to “This study, together with future work, may guide the future identification and development of therapeutics for NF1-associated tumors.”
Minor comments:
Is Benjamini-Hochberg. Correct throughout the manuscript.
We have revised the manuscript with the correct spelling.
Data is a plural noun so it should be followed by plural verbs. Check throughout the manuscript (see line 96, 108 as an example)
This has been corrected in the revision.
Line 37: I wouldn’t define NF1 a neuropathic syndrome. Rather is commonly defined as a neurocutaneous syndrome or one of the phakomatoses or RASopathies.
We have changed the description from “neuropathic syndrome” to “RASopathy.”
In supplementary figure S2, the panel B (ii) is cut
We have updated the figure file to have the correct un-cut example plot.
Reference [39] pages are missing
The citation for reference 39 has been fixed.
Reviewer 3 Report
This is a very well written, structured and scientifically sound manuscript.The authors used statistical and machine learning methods to analyze genomic data of 77 different subtypes of NF1 tumors to identify key signaling pathways and candidate genes which could be possible target of pharmacotherapy. Certainly, the methodology is in the foreground, which is extremely interesting and can also be applied to the evaluation of other genetic databases.
New molecular methods (e.g. next generation sequencing, ect.) generate a large amount of data. Identifying important genes or signalling cascades from this data is a difficult task. In this case, such integrative analysis would be a desirable and helpful tool and its findings could serve as a basis for further fundamental analyses (e.g. in vitro models) of the results obtained.
Author Response
Dear Reviewer 3,
We thank you for taking the time to provide feedback for this manuscript and your encouragement of our approach in this study.
Comments and Suggestions for Authors
This is a very well written, structured and scientifically sound manuscript.The authors used statistical and machine learning methods to analyze genomic data of 77 different subtypes of NF1 tumors to identify key signaling pathways and candidate genes which could be possible target of pharmacotherapy. Certainly, the methodology is in the foreground, which is extremely interesting and can also be applied to the evaluation of other genetic databases.
New molecular methods (e.g. next generation sequencing, ect.) generate a large amount of data. Identifying important genes or signalling cascades from this data is a difficult task. In this case, such integrative analysis would be a desirable and helpful tool and its findings could serve as a basis for further fundamental analyses (e.g. in vitro models) of the results obtained.
Round 2
Reviewer 1 Report
I thank the authors very much for having considered my suggestions, modifying the manuscript in accordance with them. Although I still have concerns about dataset, this is a well conducted study and, I hope, a good starting point for future studies.